# Working as a Healthcare Professional at Island Primary Care: An Exploratory Qualitative Study on the Cyclades Islands, Greece

**DOI:** 10.3390/healthcare12090882

**Published:** 2024-04-24

**Authors:** Anna Maria Kefala, Areti Triantafyllou, Emmanouil K. Symvoulakis, Eleni-Margarita Tzouganatou, Nikolaos Kapellas, Emmanouil Smyrnakis

**Affiliations:** 1Emergency Department, Naxos General Hospital-Health Centre, 84300 Naxos, Greece; kefalaannamaria7@gmail.com; 23rd Clinic of Internal Medicine, Papageorgiou Hospital, School of Medicine, Faculty of Health Sciences, Aristotle University of Thessaloniki, 54124 Thessaloniki, Greece; artriant@auth.gr; 3Clinic of Social and Family Medicine, Department of Social Medicine, School of Medicine, University of Crete, 70013 Heraklion, Greece; 4Clinic of Internal Medicine, Naxos General Hospital-Health Centre, 84300 Naxos, Greece; elmatzougan@gmail.com; 5Multipurpose Regional Medical Centre of Donousa, 84300 Donousa, Greece; kapellasn@gmail.com; 6Laboratory of Primary Health Care, General Practice and Health Services Research, Faculty of Health Sciences, School of Medicine, Aristotle University of Thessaloniki, 54124 Thessaloniki, Greece

**Keywords:** island primary care, rural areas, Greek islands, primary healthcare, healthcare professionals, motivations

## Abstract

Improving the quality of and access to healthcare services in rural areas is fundamental to developing sustainable healthcare systems. This research aims to explore the motivations of healthcare professionals to work and settle in rural island areas of Greece with limited access to secondary and tertiary care. The study suggests practical ways to encourage self-motivation and attract more health workers in rural areas. An exploratory qualitative research approach was employed, involving semi-structured interviews with 16 healthcare professionals working in primary-care units that lack direct hospital or hospital–health centre access. The research was conducted specifically in the rural islands of the Cyclades. Thematic analysis was conducted to identify common themes and unique insights from the participants. The analysis revealed three thematic categories. Τhe «attraction» thematic was influenced by personal factors, random selection, origin, accommodation factors, professional factors, and obligatoriness. The «recruitment» thematic was associated with understaffing, special care issues, an unstable working environment, educational and organisational aspects, and an insular lifestyle. The thematic of «retention» highlighted personal issues, accommodation difficulties, economic and work-related issues, and unique challenges posed by an insular lifestyle. This research provides valuable insights into the motivations that drive healthcare professionals to settle, work, and remain in remote island units, as well as the challenges they encounter in making this decision. The study proposes strategies to motivate and attract more healthcare professionals to rural areas. These findings should be considered when formulating or reviewing primary healthcare empowerment policies to ensure equitable healthcare access for all individuals.

## 1. Introduction

### 1.1. Background

The proper and efficient functioning of a health system requires, in addition to being well organised, a workforce with strong work motivation and high job satisfaction [1]. Health indicators are, therefore, very closely linked to healthcare providers, and the unequal distribution of health personnel in urban and rural areas negatively affects the provision of healthcare in such areas [2].

Work motivation in the health sector, particularly in health workers in decentralised areas, operates at the individual, community, and organisational levels. Health workers in primary healthcare (PHC) facilities in rural areas are motivated by altruistic instincts, a sense of responsibility, and a desire for respect and recognition of their position within a community, but also by salary, residency, and career development incentives [3,4]. In Greece, since 1983 and the establishment of the National Health System (NHS), several incentives have been introduced to mobilise healthcare workers in rural areas [5,6,7]. The incentives that have been granted to date are mainly financial and have not yet led to the desired effect.

According to the World Health Organization and the World Bank, almost half of the world’s population resides in rural areas and is served by only 38% of nursing staff and 24% of medical staff [8,9]. In Greece, according to the latest statistics from the World Bank, 19.96% of the total population lives in rural areas [10]. Although the island regions are adequately covered by healthcare facilities and there is a legal framework that mobilises at least medical personnel to them, there are still not enough human resources to cover healthcare needs and services [11]. In particular, 13.1% of rural and island residents of Greece reported unmet health needs due to distance or transport, while the European Union average was 4.5% [12]. It is, therefore, necessary to ensure equity in healthcare provision and responsiveness to the health needs of the population in rural areas, identify the shortages, and problems in healthcare units in rural areas, and design interventions that can attract health personnel with strong work incentives to remain in these structures.

A thorough literature review revealed no quantitative or qualitative study that focuses on the motivations of healthcare providers, both nurses, doctors, and other healthcare personnel to work and stay in rural Greek islands without access to secondary care. Only three quantitative studies that focused on the motivation of attraction and retention in remote and isolated health services involved solely the medical population. One was conducted in the Peloponnese Region, another in the Greek islands and areas with an increased tourist population, and the last one was held at the hospitals of Patmos, Dikaia, and Orestiada, at the hospital of Didymoteicho, and at the hospitals of Leros and Kalymnos. All three studies highlighted the unequal distribution of human resources in urban and remote areas in relation to the size of the population in these areas, the need to introduce more effective incentives, and the importance of conducting similar studies in Greece [4,6,7]. However, these studies do not fully explore the contextual factors that influence the motivation and retention of PHC workers in rural facilities of the Greek islands, a gap that this study will attempt to bridge in order to find ways of attracting more healthcare professionals in rural areas.

### 1.2. Aim

This exploratory qualitative study aimed to identify and understand the contextual issues that affect rural practice in areas with limited access to secondary and tertiary care and suggest practical ways of encouraging self-motivation among healthcare professionals in order to attract more healthcare workers to rural island areas.

## 2. Materials and Methods

This exploratory qualitative study was conducted according to the Standards for Reporting Qualitative Research (SRQR) guidelines [13].

### 2.1. Study Design

An exploratory qualitative research approach was performed in order to comprehend the factors shaping the motivation and retention of PHC workers in rural health units.

This study focuses on the islands of the Cyclades with PHC units that lack direct access to specialised care units, where health workers are currently employed under non-compulsory or non-training contracts. In qualitative research, the plurality of participants, the different perspectives, and the involvement of the researcher as part of the research process provide rich data and contribute to the interpretive research process [14]. This study design is best suited to explore and describe the deeper experiences of healthcare workers living and working in such settings [15].

### 2.2. Study Setting

This study focuses on the islands of the Cyclades, an island group in the Aegean Sea, with PHC units (HC or MPRMC) with limited access to specialised care units, where health workers are currently employed under non-compulsory or non-training contracts. 

There are 33 islands in the island complex of the Cyclades, and 21 of them met the criteria for inclusion in the study. Of these, 6 have a health centre (HC), 14 have a multipurpose regional medical centre (MPRMC), and 1 has both an HC and an MPRMC [16,17,18].

Based on the latest report of the 2nd Health Region and the Ministry of Health for 2021, the above structures employed 28 doctors, 82 nurses, and 58 other healthcare workers [16].

During the study, A.M.K., E.-M.T., and N.K. were working as clinicians in rural units of the Cyclades Islands, and their experience was a motivation for conducting the research. They remained objective throughout the study, and their experience did not interfere with any step of the process. A.T., E.K.S., and E.S. were working as professors and clinicians in tertiary hospitals and universities. 

### 2.3. Study Population, Sampling, and Recruitment of Participants

The study sample consisted of diverse healthcare workers working at the time of the survey in healthcare facilities of island areas without a hospital or hospital–health centre access, in the prefecture of Cyclades. Healthcare workers of different specialities, contracts, genders, ages, and origins were involved to ensure the diversity of experiences explored. Healthcare practitioners in a training or compulsory working context were excluded from the study.

Non-random purposive sampling was followed, based on which, the researcher selected the most appropriate participants who could provide rich information to answer the study question in depth [19]. There were neither refusals nor drop-outs. The sample was determined based on the theoretical saturation effect and was completed when data collection and analysis could not provide additional evidence to formulate a new theory [20] (p. 1249).

### 2.4. Data Collection

A total of 16 in-depth interviews (IDIs) were conducted in Greek either face-to-face or by telephone for data collection. They ranged from 20 to 35 min and were completed over two months, during November and December of 2022. Most of the interviews, eleven out of sixteen, were conducted by telephone due to unstable weather conditions leading to difficulties in transportation. High-quality audio recordings of the interviews were made after the participants’ informed consent. The interviews were translated into English prior to publication by A.M.K and N.K. and the translation was approved by all authors.

Semi-structured interviews were used. Eleven questions were formulated to cover the topic, which were structured in such a way as to promote discussion. A semi-structured interview is introduced by open questions that help the participants to unfold their stories. Then, questions around the research topic are made in a way that promotes dialogue [14]. The questions were carefully chosen to avoid guided answers and were approved by the research team. Prompting, clarifying, supplementary, interpretive, and implicit prompting questions were used.

In addition to the interviews, an information sheet was completed to record some demographic data and information about the conditions prevailing in each PHC facility.

### 2.5. Data Analysis

The thematic analysis technique was used to analyse the data [21]. Based on this approach, the following steps were followed. First, the recorded interviews were listened to and transcribed, with detailed transcription without omission of details, pauses, or comments. The quotes relevant to the research questions were then identified and collated by two researchers independently. The coded data were cross-checked and then discussed with the whole research team to resolve any coding discrepancies. This interactive, analytical, and iterative process involved re-categorizing and combining codes into themes.

## 3. Results

### 3.1. Demographic Characteristics

The demographic characteristics are provided in Table 1.

### 3.2. Facility Characteristics

The survey included health workers working during the study period in seven PHC units, four HCs, and three MPRMHs, in rural island areas of the Island Complex of the Cyclades. The names of the facilities and population data are not reported to ensure the anonymity of the participants. For patient examination, most of the HCs were equipped with an electrocardiograph, monitor/defibrillator, ultrasound and radiology machine, and biochemical and blood gas analyser. They also had a dental machine and the ability to perform Pap-smear. Most of the MPRMHs were equipped with an electrocardiograph, monitor/defibrillator, haematology analyser, biochemistry analyser, and radiology machine.

### 3.3. General Results

The analysis of the transcribed interviews revealed three thematic categories, with subcategories that influence the attraction, recruitment, and retention of healthcare professionals in primary healthcare facilities in rural island areas of Greece and are presented in Figure 1.

#### 3.3.1. Attraction

Personal factors: showed that the settlement of health workers in rural and remote islands of the Cyclades was associated with family and emotional reasons, as well as with the need to meet personal aspirations such as providing healthcare to underprivileged communities and the need for a calmer pace of life and work. Reference was also made to the issue of the difficulty of health workers settling in a rural place due to problems with the partner’s professional rehabilitation or opportunities for children’s education and recreation.


*I worked for years as a private doctor but having a dream growing up to live a more peaceful life on an island and to support the public sector and the PHC with my work, is why I came to this island and Health Centre.*
(IDI 15)

Random selection: Health professionals reported an unintended choice of work structure.


*The choice of this particular island and structure was made randomly. I really wanted to work in the public sector and get credits. Actually, I also wanted to try something different.*
(IDI 4)

Origin: As also seen in Table 1, six of the participants returned to work in their place of origin and all expressed a desire to remain on the island, while four of them expressed a desire to remain on the island and at their jobs.


*I prefer this place because I am from here, my family is here... It is not easy to live on an island and the truth is that if I were not from here, I don’t know if I would have chosen to come to such a distant place, so far away from a hospital or organized health structures.*
(IDI 11)

Accommodation: Housing was discussed as a matter of discouragement for health workers from settling on rural and remote islands.


*We have a problem with houses. A health worker may want to come and have a position but there is no house to inhabit.*
(IDI 13)

Professional factors: The theme of attraction was also associated with professional reasons, such as previous experience in the facility, the opportunity to practice and improve clinical skills, and opportunities for career advancement either through grading or tenure. Professional considerations were also mentioned as having a negative impact on attracting staff to rural areas.


*Working at a big hospital during the three years of the pandemic of COVID-19 were crusial for my choice to move in a small island with a calmer way of life. I had previously worked at this island and really liked it. You have the opportunity to practice your clinical skills and find different ways for dealing with cases, when lacking diagnostic and therapeutic tools.*
(IDI 7)

Obligatoriness: Working in a rural PHC unit in the form of a rural service of medical specialists was proposed by two health workers as a means of installing staff in PHC structures on rural islands.


*One way to have health coverage in these areas is to have rotating health workers. To create a form of rural specialty doctors’ rotation. Any new specialist on completion of residency could, for example, serve in a rural area for six months. In this way, the border areas would be continuously covered.*
(IDI 10)

#### 3.3.2. Recruitment

Understaffing: All healthcare professionals highlighted the issue of staff shortages leading to several problems. The lack of human resources leads to overwork, constant preparedness, increased workload despite reduced attendance during the winter months, and increased sense of responsibility and work stress. Some of the participants also highlighted the lack of a health team, a condition that makes it difficult to operate a lean structure, complicates case management, and causes problems of cooperation between health workers. Reference was also made to the need to attract a team of health workers to work instead of advertising individual posts as an incentive to increase staffing in lean structures.


*Difficulties evidently exist. There is no replacement when I need some annual leave. There is no concept of duties. We do many things beyond our duties either administrative or ordering medication etc. I do not focus solely on my nursing duties.*
(IDI 1)

Special care issues: Although a few health workers reported an improvement in diagnostic tools in the facility where they work compared to their previous experience, the majority reported reduced possibilities of specialised care. The professionals focused on the issues of limited diagnostic tools, lack of qualified staff, and referral and transfer issues for severely ill patients who need further investigation and treatment. Considering these, the health workers suggested the improvement of the diagnostic capacity of the facilities and the need for rural and emergency-medicine training for all staff before working in a rural setting. Some health professionals referred to the possibility of providing specialised healthcare through telemedicine but stressed the need to better inform the population about its importance and use and the need to serve more specialities and emergencies.


*I’ve been in the MPRMC for six years. The first three were very difficult. There was no equipment, or some were there but they were malfunctioning. In the last few years, with maintenance and the use of a new building, we have some microbiology equipment providing a first assessment and better equipment in general. We used to refer every patient to a bigger structure in the past, but now, at least we can have first assessment.*
(IDI 1)

Unstable working environment: Most participants highlighted the issue of the lack of stability of health staff in each facility, which creates both issues of collaboration within the facility and issues of trust between health workers and the community.


*Also, things in the working environment change every year. There is instability. Because the health workers, and especially the rural doctors, change every year.*
(IDI 9)

Scientific/educational aspects: An issue that was raised by the majority of the participants and which falls within both the theme of recruitment and retention is that of scientific information and continuing education. The lack of opportunity to attend scientific conferences and seminars was mentioned as a challenge, both by doctors and nurses.


*Workwise, I will say that there really is no progression when you work on an island. The conferences that take place, at least for us nurses, are all in bigger islands and cities, so with the salary we get, we can’t go. There are not many online conferences or seminars… And they should be free for those of us who are in remote health units.*
(IDI 4)

Organisational aspects: The main organisational issue raised by all health staff of the facilities was the lack of an organised patient transfer procedure. A coordinated action plan by the National Medical Emergency Service (EKAB) and the secondary or tertiary hospitals that accept critically ill patients, is not available. This increases both the work stress of health workers working in rural health units and the health risk of patients visiting these structures. The health workers reported difficult communication with the EKAB, while some, despite acknowledging a lack of organization, were satisfied with its response. Difficulties in linking up with other structures, mainly due to the latter’s ignorance of the conditions prevailing in poor facilities, were also reported. In relation to the above challenges, most of the participants suggested the creation of an ambulatory coordination team to assist with the job being done by the staff in rural units.


*This is the worst part of the job because not all colleagues in the hospitals we call are cooperative. Many don’t want to accept the patient’s referral and don’t really understand the limited options of treatment we can provide in a remote area… We are forced to make phone calls repeatedly until we get an acceptance, which is a time-consuming action that interrupts the therapeutic procedure of a severely ill patient.*
(IDI 7)

Insular lifestyle: Living on an island seems to influence retention in various aspects. Even when the necessary equipment is available, maintenance is difficult, there are delays in the provision of medical equipment, the need for care increases during the summer season, and weather conditions create issues in cases where both waterborne and air ambulance services are needed. At the same time, the impact of a closed community on the way a health unit works was raised, as well as the big percentage of the older population, which accompanies an increased prevalence of chronic diseases and, hence, a greater need for healthcare provision.


*During the summer, the workload increases because more people visit the island, but the staff remains the same and is insufficient to cover the health needs.*
(IDI 7)

#### 3.3.3. Retention

Personal issues: The interviews clearly revealed that family reasons, achieving personal and work goals, and the matter of special care provision are personal issues that influence the retention of health workers at health facilities in remote islands.


*Key specialties are missing from the island. That said, having a family and not having a pediatrician on the island is extremely difficult.*
(IDI 3)

Accommodation: Accommodation issues were mentioned by all participants. Local health workers considered the availability of housing as an incentive for staying on the island and at the health unit, while others referred to the increased cost of accommodation, lack of accommodation infrastructure, and the need for an incentive for accommodation either financially or as a provision of space.


*I would say that accommodation is difficult though, as the island is a tourist destination and it is not easy to find a place to stay, especially if you come with your family.*
(IDI 1)

Economic issues: The mismatch of living and travel costs versus income, and the mismatch of income with the increased responsibility and preparedness of health workers on a rural island, is one of the major issues emerging from the survey that affects the stay of health workers in remote areas.


*Expensive living. All small islands are expensive because there is no competition to bring prices down. There are no big supermarkets. This makes living here very difficult.*
(IDI 4)

Work-related factors: The interviews revealed the issues of increased workload and sense of responsibility experienced by workers in rural healthcare facilities. Rewarding feelings, such as moral satisfaction and a sense of contribution to the community and PHC, also emerged as work-related factors.


*I have a permanent position at a faculty on my island and I am close to my family. I am happy to be able to offer to the people who live here through my job.*
(IDI 8)

Insular lifestyle: Weather challenges, difficulties in moving, difficulty in accessing goods, reduced recreational and educational options, as well as isolation and feelings of loneliness, are issues that affect health workers staying in remote areas. Others, mainly local health workers and health workers who choose a more relaxed lifestyle, do not associate their choice with the above-mentioned challenges.


*I am thinking of not staying but not because of the job. What makes it difficult for me to stay is the distance and isolation. The fact is that we don’t have a plane, and we can’t leave at any time. So, I have thought about leaving the island to live closer to my family and friends. But I would still live in a small place.*
(IDI 13)

## 4. Discussion

Staffing healthcare units with human resources and having the best performance from health professionals are essential ingredients for the overall performance of the healthcare system. Many factors either facilitate or hinder the achievement of this goal [22]. This research, through the stories of PHC professionals, highlights the various challenges they face when deciding to live and work in rural areas, but also reveals the advantages and opportunities that accompany this decision (Figure 2). The study also suggests ways of strengthening the provision of healthcare to the population of remote areas, mainly through improving the working and living conditions, attracting more health workers, and ensuring equal healthcare access for all citizens.

Attraction, recruitment and retention of health workers on rural islands

The most important issues associated with settling and retaining on rural islands are rural origin, previous experience, living conditions, working conditions, scientific limitations, and personal reasons.

Despite its appearance in the study, the economic issue is not prominent and emerges mainly as related to working conditions and its mismatch with job responsibility and workload. This observation is confirmed by the findings of many international studies and systematic reviews, as well as by one of the three studies related to this research in Greece [2,7,22,23,24]. 

In addition, it was obvious that some of the health workers returned to work in a health facility where they had previously worked. Previous experience in a health facility in a rural area has been strongly associated with the decision of healthcare personnel to work in a similar facility. The matter of obligatoriness was associated only when combined with strong financial incentives [9].

Rural origin acts as a factor of attraction and retention, which aligns with the existing international literature. The need to offer healthcare services in the place of origin, the close family ties, and, to a lesser extent, the existence of a residence are the factors that seem to be the main motivators. Based on this finding, countries such as Australia have found places to attract future local health workers, such as the UYDF SS program, which provides scholarships to young students from rural areas who wish to pursue a health profession, to increase the likelihood of returning to work in their home country [25].

Living conditions, especially the high cost of food and the lack of housing at a price commensurate with the salaries of health professionals, are major obstacles to the mobilisation of health workers on small and rural islands. In the literature, this finding occurs mainly in rural areas in developing countries [2].

All participants acknowledged the issue of understaffing of health facilities in remote areas and highlighted the work stress resulting from the need to take on increased responsibilities. This, combined with shortages of medical equipment, diagnostic tools, and specialist doctors, and the problem of the unstable working environment, discourages health workers from working and remaining in health units in remote areas [26]. An important research result that enriches the knowledge about the challenges that arise when working in rural areas in Greece is the issue of communication and transfer of severely ill patients at secondary and tertiary hospitals. In many cases, hospitals are poorly informed about the prevailing difficulties of healthcare provision in remote areas and the stress accompanying case management, an issue that often leads to delays in treatment. A study carried out in Canada highlights the vital importance of good communication between structures in providing safe and good-quality healthcare to patients in rural areas [27].

Opportunities to enhance scientific knowledge allow health professionals to stay up-to-date and promote their career development. The scientific constraints that accompany working in PHC facilities on islands are a disincentive to settling and staying, as confirmed by other research in Greece concerning health professionals working in remote areas [4,7]. The possibility of continuing education through postgraduate programs, conferences, and seminars contributes to the scientific awareness of health professionals and is very important for their job performance [28].

Finally, the reasons influencing the decision of health professionals to both settle, work, and remain in isolated Greek island regions, as well as in small and isolated areas around the world, are the feeling of isolation, the distance from their close family and friends, and the limited recreational and educational options [23]. Of course, the quality of life, the close ties developed with the community, and the sense of offering to people in need living in rural areas, often compensate for the existing challenges [1,3].

Proposals for attracting health workers to rural island units

Based on the data obtained from the interviews and taking into account the Greek and international literature review, and the WHO’s detailed recommendations, the following incentives to strengthen rural structures with health personnel are proposed [9,25,29,30,31]. In order to attract more healthcare professionals in rural settings, it is important to create strong financial incentives along with improving housing and living conditions and ensuring a safe working environment with improved diagnostic and therapeutic tools. A safe working environment will be enhanced by providing emergency-medicine training opportunities for all rural health workers by formulating a referral coordination team and recruiting a healthcare team. Increasing educational leaves, providing promotion and continuing education opportunities, and empowering healthcare team spirit are also proposed. Providing study opportunities in the form of scholarships for students of rural origin and rural training and orientation for health students are among the strongest incentives supported by the literature review. Using NGOs to help improve island-lifestyle opportunities is a measure that increases the possibility of healthcare professionals remaining in a rural structure.

Strengths and limitations

To the best of our knowledge, the present exploratory qualitative analysis is the first demonstration of healthcare professionals’ motivational and retention factors in PHC facilities on rural islands of Greece that have limited access to secondary and tertiary care.

The main limitation of our study is the small sample of healthcare workers working in healthcare facilities on rural islands of the island group of the Cyclades and, therefore, the inevitable generalisation of the results. Despite the small sample size, the information collected and the emerging themes fully answer the main research question and sub-questions and fill the research gap on this topic in the Greek health sector. Moreover, a single data-collection tool was used, that of the interview, thus eliminating the possibility of triangulating the data through a combination of qualitative and quantitative approaches.

## 5. Conclusions

The present study is the first to interview health professionals, doctors, nurses, and other health personnel and explore the subject of their decision to live and work in healthcare facilities in small Greek islands. Their beliefs could facilitate ways of improving attraction, recruitment, and retention in remote areas. As job satisfaction is linked to job performance and better healthcare delivery, providing health workers with high incentives to work and stay in remote areas also ensures equal access to healthcare for all citizens and improves the overall performance of the health system in Greece.

Although the results of a single survey may seem small, the widespread engagement of the global community and WHO with the issue of health worker disparity in rural areas, and the right of all citizens to equal opportunities for care, highlights the need to include them when formulating or reviewing policies to strengthen PHC in Greece and increase the sustainability of the health system. Finally, this research can be used as a guide for developing the new field of Island Medicine, as it sheds light on unknown aspects of the activities of healthcare providers in island regions.

## Figures and Tables

**Figure 1 healthcare-12-00882-f001:**
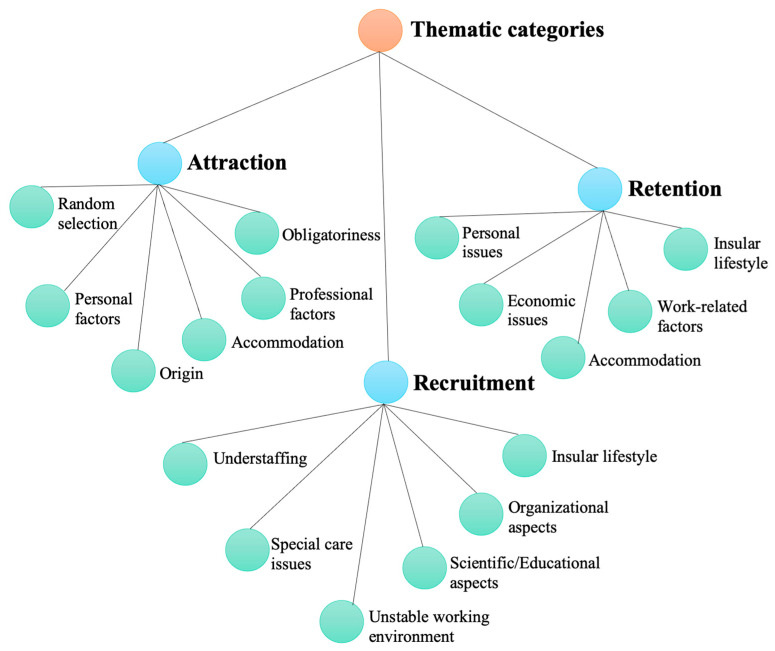
Thematic analysis.

**Figure 2 healthcare-12-00882-f002:**
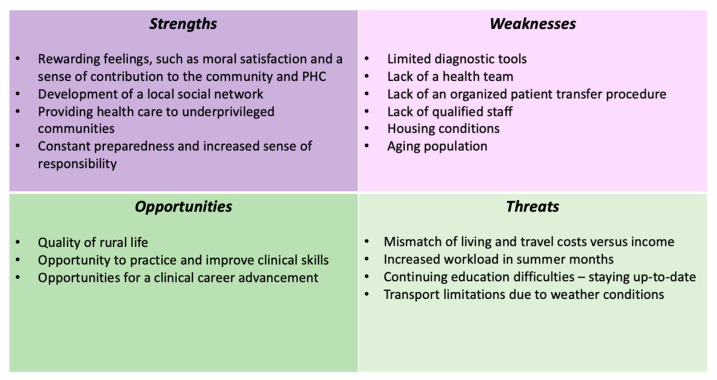
Swot analysis for deciding to follow a career at a rural primary-care unit.

**Table 1 healthcare-12-00882-t001:** Demographic characteristics of participants.

IDI	Gender	Age	Native	Marital Status	Kids	Level of Education	Type of Structure	Specialty	Years Working in the Structure	Contract Type	Desire to Remain in the Structure	Desire to Remain on the Island
1	Female	30	Yes	Single	-	Bachelor	MPRMC	Nurse	6	Permanent	Yes	Yes
2	Female	49	No	Single	1	Bachelor	MPRMC	GP	2	Permanent	No	Yes
3	Female	33	No	Single	-	Master	HC	Nurse	3	Temporary	No	No
4	Male	42	No	Divorced	2	Bachelor	HC	Nurse	12	Permanent	No	No
5	Female	45	No	Divorced	1	Doctoral	HC	Cardiologist	2	Permanent	Yes	Yes
6	Female	53	No	Married	1	Bachelor	MPRMC	Nurse	6	Permanent	No	Yes
7	Female	33	No	Single	-	Bachelor	HC	Internist	2	Temporary	Yes	Yes
8	Female	29	Yes	Married	1	Master	HC	Nurse	6	Permanent	Yes	Yes
9	Female	39	Yes	Married	2	Bachelor	HC	Nurse	17	Permanent	Yes	Yes
10	Male	52	No	Married	-	Master	HC	Cardiologist	1	Permanent	No	No
11	Female	45	Yes	Married	2	Bachelor	HC	Nurse	23	Permanent	No	Yes
12	Male	43	Yes	Single	-	Bachelor	HC	Orthopaedic	7	Temporary	No	Yes
13	Female	42	Yes	Married	-	Master	HC	Midwife	6	Permanent	Yes	Yes
14	Female	41	No	Single	-	Bachelor	MPRMC	Nurse	9	Permanent	Yes	Yes
15	Male	64	No	Divorced	2	Bachelor	HC	Paediatrician	1	Temporary	No	No
16	Male	41	No	Married	2	Master	HC	Dentist	5	Temporary	No	No

## Data Availability

The original contributions presented in the study are included in the article. Further inquiries can be directed to the corresponding author.

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
