# Peer review of "Working as a Healthcare Professional at Island Primary Care: An Exploratory Qualitative Study on the Cyclades Islands, Greece"

_healthcare, 2024, doi:10.3390/healthcare12090882_

Round 1

Reviewer 1 Report

Comments and Suggestions for Authors

Title of Manuscript:

I thank the author for putting together “Working as a healthcare professional at island primary care: an exploratory qualitative study in Cyclades Islands, Greece”

Comments: the findings of this study would be useful in guiding the recruitment and retention of critical medical staff in rural areas in general and in Cyclades Islands in particular. This is a good paper and well written.

Abstract: The abstract is well written. It contains all the relevant information needed to describe what was done and found.

Introduction

The introduction provides adequate background and rationale to inform the reader about what to expect.

The objectives of the study is also clear

Materials and Methods

Though the authors provided information on the duration of study the fail to inform the reader when the data was collected.

The authors mentioned that 16 IDI were collected using various methods (face-to-face/telephone). It is imported to inform reader’s reasons for or what circumstances under which each of these data collection methods was applied.

General results

The presentation of the results is okay. However, some of the quotes does not adequate reflect the write up. For example, line 194-198: Professional factors. It is difficult to reconcile the citation with the write up. “I had worked at another rural island before, and I just wanted to offer. So that's why I chose 199 it. (IDI 5)” How does this support the professional reasons provided or improve the opportunity to practice and improve clinical skills.

Discussion

In general, the discussion is rich and provide several insight and deep understanding of the various reason why it is challenging for medical staff to stay and work in rural areas in Greece.

Reviewer 2 Report

Comments and Suggestions for Authors

Thank you very much for the opportunity to read this paper, which qualitatively explores the experiences and attitudes of doctors working in rural/remote Greek islands. This is a lovely paper, well backgrounded, the findings are well structured and have a nice sense of participant voice, and there are clear connections to broader global knowledge on rural/remote healthcare workforce and strong recommendations made. 

One thing I would like to see in the methodology section is a positionality statement/reflection from the authors, about their own positions - are they clinicians, or social scientists? Have they worked in rural/remote areas, or come from/grew up in those areas, themselves? How have they reflected on who they are and what perspectives they bring to this topic, and how did they use this, or try to mitigate against it as needed, in the research?

Another minor addition/clarification in methods - were the interviews conducted in English, as the data is presented in English here? Or were they conducted in Greek, and have been translated to English? When did translation occur - pre-analysis, or just pre-publication? Who has done the translation - was it a member of the research team? Was it checked/agreed on by all members of the team?

Line 161 - referencing error

Reviewer 3 Report

Comments and Suggestions for Authors

Thank you for the opportunity to review this important work which has the potential to inform strategies for appropriate recruitment and retention of rural primary health workforce, not just only in rural Greece but other similar countries. The manuscript is easy to read and the SWOT analysis is easy to understand. I wonder if the authors may wish to elaborate on how the 11 questions used for the interviews were developed/ derived and/or validated. I look forward to reading and sharing this manuscript when published. 
